# COMPETENCE-BASED ANALYSIS OF LANGUAGE MODELS

## ABSTRACT

Despite the recent successes of large language models (LLMs), little is known regarding the representations of linguistic structure they learn during pretraining, which can lead to unexpected behaviors in response to prompt variation or distribution shift. To better understand these models and behaviors, we introduce a general model analysis framework to study LLMs with respect to their representation and use of human-interpretable linguistic properties. Our framework, CALM (Competence-based Analysis of Language Models), is designed to investigate LLM competence in the context of specific tasks by intervening on models' internal representations of different linguistic properties using causal probing, and measuring models' alignment under these interventions with a given ground-truth causal model of the task. We also develop a new approach for performing causal probing interventions using gradient-based adversarial attacks, which can target a broader range of properties and representations than prior techniques. Finally, we carry out a case study of CALM using these interventions to analyze and compare LLM competence across a variety of lexical inference tasks, showing that CALM can be used to explain and predict behaviors across these tasks.

## 1 INTRODUCTION

The rise of large, pretrained neural language models (LLMs) has led to rapid progress in a wide variety of natural language processing tasks Brown et al. (2020); Chowdhery et al. (2022); Dubey et al. (2024). However, these models can also be quite sensitive to minor changes in input prompts Elazar et al. (2021a); Moradi & Samwald (2021); Mizrahi et al. (2024) and fail to generalize outside their training or fine-tuning distribution Wang et al. (2023a); Yang et al. (2023). It is usually unclear where these limitations come from, as LLM task performance is generally studied using only "black box" behavioral analysis, in which case one can only detect limitations that are adequately represented by the benchmark, which cannot cover every possible limitation using a finite dataset Raji et al. (2021); Siska et al. (2024). Understanding the means by which these models can perform as well as they do while exhibiting such limitations is a key question in the science of LLM interpretation and analysis Bereska & Gavves (2024), and is likely necessary in enabling robust, trustworthy, and socially-responsible LLM-enabled applications Shin (2021); Liao & Vaughan (2023); Zou et al. (2023); Bereska & Gavves (2024).

We approach this question in terms of *competence*, drawing on the traditional competence-performance distinction in linguistic theory (see Section 2.1) to motivate the study of LLMs in terms of their underlying representation of language. We define LLM competence in the context a given linguistic task as the alignment between the ground-truth causal structure of the task and the LLM's latent representation of the task's structure, measured by intervening on the LLM's representation of task-causal or non-causal properties and observing how its behavior changes in response. While such representations are not directly observable, we take inspiration from *causal probing*, which damages LLMs' latent representations of linguistic properties using causal interventions to study how these representations contribute to their behavior Elazar et al. (2021b); Lasri et al. (2022). We introduce a general framework, CALM (for Competence-based Analysis of Language Models), to study the competence of LLMs using causal probing and define the first quantitative measure of LLM competence.

While CALM can be instantiated using a variety of existing causal probing interventions (e.g., Ravfogel et al., 2020; 2022b;a; Shao et al., 2022; Belrose et al., 2024), we develop a new intervention methodology for damaging LLM representations using gradient-based adversarial attacks against structural probes, extending causal probing to arbitrarily-encoded representations of relational properties and thereby enabling the investigation of new questions in language model interpretation. We carry out a case study of CALM using two well-studied LLMs by implementing interventions as GBIs in order to measure and compare these LLMs' competence across 14 lexical inference tasks, showing that CALM can indeed explain and predict important patterns in behavior across these tasks by distinguishing between models' use of causal and spurious properties.

Our primary contributions are as follows:

1. We introduce CALM, a general interpretability framework for studying LLM competence using causal probing.

2. We provide a causal formulation of linguistic competence in the context of LLMs, using CALM to define the first quantitative measure of LLM competence.

3. We establish a gradient-based intervention strategy for causal probing, which directly addresses multiple limitations of prior methodologies.

4. We discuss multiple novel applications enabled by CALM for understanding the representation and behaviors of LLMs.

5. We implement a preliminary case study of CALM using gradient-based interventions, demonstrating its utility in explaining and predicting LLM behaviors across several lexical inference tasks.

## 2 COMPETENCE-BASED ANALYSIS OF LANGUAGE MODELS

### 2.1 LINGUISTIC COMPETENCE

Linguistic competence is generally understood as the ability to utilize one's knowledge of a language in producing and understanding utterances in that language, and is typically defined in contrast with linguistic performance, which is speakers' actual use of their language in practice, considered independently of the underlying knowledge that supports it Marconi (2020).[1] Given a linguistic task, we may understand competence in terms of the underlying linguistic knowledge that one draws upon to perform the task. If fluent human speakers rely on (implicit or explicit) knowledge of the same set of linguistic properties to perform a given task, then we may understand their performance of this task as being causally determined by these properties, and invariant to other properties. For example, if we consider the two utterances "the chicken crosses the road" and "the chickens cross the road", the grammatical number of the subject (i.e., singular and plural, respectively) determines whether the verb "(to) cross" should be conjugated as "crosses" or "cross". As English (root) verb conjugation always depends on the grammatical number of the subject, grammatical number may be regarded as having a causal role in the task of English verb conjugation, so we may understand fluent English speakers' (usually implicit) mental representation of verb tense as having a causal role in their behavior. In this work, we focus on *lexicosemantic competence*, the ability to utilize knowledge of word meaning relationships in performing tasks such as lexical inference Marconi (1997; 2020).

While the study of human competence has a rich history in linguistics, there is currently no generally accepted framework for studying LLM competence Mahowald et al. (2023); Pavlick (2023). Our primary goal in this work is to define and test a general empirical analysis framework for interpreting and measuring LLM competence, as outlined in the following section.

---

[1]There has been significant debate in linguistics and the philosophy of language regarding the precise definition and nature of competence Lyons (1977); Newmeyer (2001); Sag & Wasow (2011); Marconi (2020). However, the formalization of competence provided in this work is sufficiently general to incorporate most notions of competence, which may be flexibly specified by instantiating CALM in different ways.

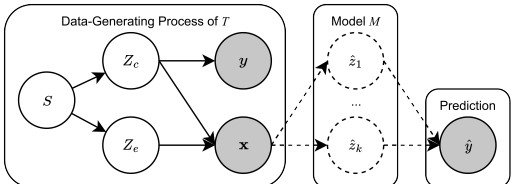 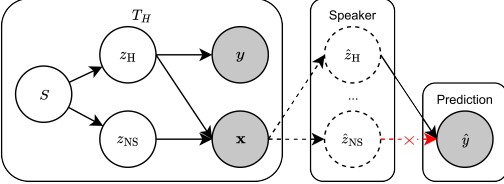

Figure 1: Structural causal model (SCM) of task $\mathcal{T}$'s data-generating process and how it may be performed by model $M$. Shaded and white nodes denote observed and unobserved variables, respectively. In CALM, the goal is to determine which representations $Z_j = z_j$ are causally implicated in $M$'s predictions $\hat{\mathbf{y}}$.

Figure 2: SCM of a competent English speaker on the hypernym prediction task. Shaded and white nodes denote observed and unobserved variables, respectively.

## 2.2 CALM FRAMEWORK

In order to make the study of competence tractable in the context of LLMs, we introduce the CALM (Competence-based Analysis of Language Models) framework, which describes an LLM's competence with respect to a given linguistic task in terms of its latent representation of the causal structure of the task.

**Task Structure** Formally, given supervised task $\mathcal{T} \sim P(\mathcal{X}, \mathcal{Y})$ where the goal is to correctly predict $\mathbf{y} \in \mathcal{Y}$ given $\mathbf{x} \in \mathcal{X}$, and a collection of latent properties $\mathbf{Z} = \{Z_j\}_{j=1}^m$ that are (potentially) involved in generating $\mathbf{x}$, we formulate the causal structure of $\mathcal{T}$ in terms of the data-generating process

$$\mathbf{x} \sim \Pr(\mathbf{x}|\mathbf{Z}_c, \mathbf{Z}_e), \quad \mathbf{y} \sim P(\mathbf{y}|\mathbf{Z}_c) \tag{1}$$

where $\mathbf{Z}$ may be decomposed into $\mathbf{Z} = \mathbf{Z}_c \cup \mathbf{Z}_e, \mathbf{Z}_c \cap \mathbf{Z}_e = \varnothing$, where $\mathbf{Z}_c$ contains all properties that causally determine $\mathbf{y}$, and $\mathbf{Z}_e$ are the remaining properties that may be involved in generating $\mathbf{x}$ (cf. Ilse et al., 2021). However, there may be an unobserved confounder $S$ that produces spurious correlations between $\mathbf{y}$ and $\mathbf{Z}_e$, which, if leveraged by language model $M$ in the course of predicting $\hat{\mathbf{y}}$, can lead to unexpected failures on $\mathcal{T}$ when the spurious association is broken Pearl (2009). The structural causal model (SCM)[2] of this data-generating process is visualized on the left side of Figure 1.

For example, suppose a speaker wants to communicate that orangutans are a genus of primate. She might say "orangutans are primates" or "orangutans, a genus of apes, are primates". In both cases, the conjugation of the root verb would be "are" because it is independent of whether the subject is complemented by an appositive phrase like "a genus of apes", and this phrase does not change the grammatical number of the subject "orangutans"; so if we define $\mathcal{T}_{\text{VC}}$ as English verb conjugation, $Z_{\text{NS}}$ as the grammatical number of the subject, and $Z_{\text{AP}}$ as the presence of an appositive phrase modifying the subject, then it is clear that $Z_{\text{NS}} \in \mathbf{Z}_c$ and $z_{\text{AP}} \in \mathbf{Z}_e$. However, if we instead consider the task $\mathcal{T}_{\text{H}}$ of predicting hypernyms – for example, predicting $y$ in "orangutans are $\mathbf{y}$s", where $\mathbf{y} =$ "primate" and $\mathbf{y} =$ "ape" would both be correct answers – the causal property $Z_{\text{H}} \in \mathbf{Z}_c$ will be the hypernymy relation, and $Z_{\text{NS}} \in \mathbf{Z}_e$ (e.g., the same answers will be correct if the question is instead posed as "an orangutan is a $\mathbf{y}$"). Thus, we expect competent English speakers to be invariant to grammatical number when performing hypernym prediction (see Figure 2).

**Internal Representation** Our main concern is measuring how attributable an LLM $M$'s behavior in a given task $\mathcal{T}$ is to its representation of various properties $\mathbf{Z} = \{Z_1, ..., Z_m\}$, and how these properties correspond to the causal structure of the task. If $M$ respects the data-generating process of $\mathcal{T}$, then its behavior should be attributable only to causal properties $Z \in \mathbf{Z}_c$ (and not to environmental properties $Z \in \mathbf{Z}_e$), in which case we say that $M$ is *competent* with respect to $\mathcal{T}$ (see Figure 2). We study model $M$'s use of each property $Z_j \in \mathbf{Z}$ by performing causal interventions $\text{do}(Z_j)$ on its representation of $Z_j$ in the course of performing task $\mathcal{T}$, and measure the impact that these interventions have on its predictions.

---

[2]An SCM is a directed acyclic graph where each node represents a variable and directed edges indicate causal dependencies (see Bongers et al. 2021 for an introduction to SCMs).

## 2.3 MEASURING COMPETENCE

We evaluate the competence of $M$ with respect to task $\mathcal{T} \sim P(\mathcal{X}, \mathcal{Y})$ by measuring its causal alignment with a *competence graph* $\mathcal{G}_\mathcal{T}$, which we define as a structural causal model (SCM) of $\mathcal{T}$ with nodes corresponding to each latent variables $Z_j \in \mathbf{Z}$ and an additional node for outputs $\mathbf{y} \in \mathcal{Y}$ and directed edges denoting causal dependencies between these variables. That is, the set of causal properties $\mathbf{Z}_c$ defined by $\mathcal{G}_\mathcal{T}$ is the set of all properties $Z_j \in \mathbf{Z}$ such that there is an edge or path from $Z_j$ to $\mathbf{y}$.

To determine the extent to which $M$'s behavior is correctly explained by the causal dependencies (and lack thereof) in $\mathcal{G}_\mathcal{T}$, we measure their consistency under interventions $\mathrm{do}(\mathbf{z})$, where setting $\mathbf{z} = \{z_j\}_{j=1}^m \sim \mathrm{val}(\mathbf{Z})$ is a combination of values $Z_j = z_j \in \mathrm{val}(Z_j)$ taken by each corresponding latent variable $Z_j \in \mathbf{Z}$. For instance, under the hypernym prediction task $\mathcal{T}_H$, for input $\mathbf{x}_i =$"orangutans are $\mathbf{y}$s" and ground-truth output $\mathbf{y} =$"primate", the values taken by $\mathbf{z}_i$ would be $Z_\mathrm{H} = 1, Z_\mathrm{NS} = 1$ (where $1$ indicates the presence of hypernymy and a plural noun subject, respectively), and we might define an alternative $\mathbf{z}'$ where $Z_\mathrm{H} = 0, Z_\mathrm{NS} = 1$, under which a competent model's prediction would be expected to change with the causal variable $Z_\mathrm{H}$ (i.e., $M(\mathbf{x}|\,\mathrm{do}(\mathbf{z}')) \neq M(\mathbf{x})$).

The alignment of $M$ with $\mathcal{G}_\mathcal{T}$ is measured in terms of the similarity $S$ of their predictions under interventions $\mathrm{do}(\mathbf{z})$ given input $\mathbf{x} \sim P(\mathcal{X})$, and can be computed using a given similarity metric $S : \mathcal{Y}, \mathcal{Y} \to [0, 1]$ (e.g., equality, n-gram overlap, cosine similarity, etc.) depending on the SCM $\mathcal{G}_\mathcal{T}$ and output space $\mathcal{Y}$. That is, we define $\mathcal{C}_\mathcal{T}(M|\mathcal{G}_\mathcal{T})$ as $M$'s competence with respect to task $\mathcal{T}$ as a function of its alignment with corresponding task SCM $\mathcal{G}_\mathcal{T}$ under interventions $\mathrm{do}(\mathbf{z})$ measured by similarity metric $S$, as follows:

$$\mathcal{C}_\mathcal{T}(M|\mathcal{G}_\mathcal{T}) = \mathbb{E}_{\mathbf{x},\mathbf{z}\sim P(\mathcal{X},\mathrm{val}(\mathbf{Z}))} S\big(M(\mathbf{x}|\,\mathrm{do}(\mathbf{z})), \mathcal{G}_\mathcal{T}(\mathbf{x}|\,\mathrm{do}(\mathbf{z}))\big) \tag{2}$$

This $\mathcal{C}_\mathcal{T}(M|\mathcal{G}_\mathcal{T})$ metric (bounded by $[0, 1]$) is an adaptation of the Interchange Intervention Accuracy (IIA) metric (Geiger et al., 2022; 2023) to the context of causal probing, where instance-level interventions are replaced with concept-level interventions enabled by the gradient-based intervention methodology we introduce in Section 3. (See Appendix C.1 for a detailed comparison of our competence metric with IIA.)

## 2.4 CAUSAL PROBING

A key technical challenge in implementing CALM (and causal probing more generally) is designing an algorithm to perform causal interventions $\mathrm{do}(Z)$ that maximally damage the representation of a property $Z$ while otherwise minimally damaging representations of other properties $Z'$ Ravfogel et al. (2022b). For example, *amnesic probing* Elazar et al. (2021b) uses the INLP algorithm Ravfogel et al. (2020) to produce interventions $g_Z$ that remove all information that is linearly predictive of property $Z$ from a pre-computed set of embedding representations $\mathbf{H}$, showing that BERT makes variable use of parts-of-speech, syntactic dependencies, and named-entity types in performing masked language modeling. However, Elazar et al. (2021b) also found that, when INLP is used to remove BERT's representation of these properties in early layers, it is often able to "recover" this representation in later layers, which is likely due to BERT encoding these properties nonlinearly; and later work has found that the same "recoverability" problem persists even when linear information removal methods like INLP are kernelized Ravfogel et al. (2022b). Thus, it is necessary to develop interventions that do not require restrictive assumptions about the structure of LLMs' representations (e.g., linearity; see Vargas & Cotterell 2020), a problem which we aim to solve in the following section.

## 3 GRADIENT-BASED INTERVENTIONS

Our goal in developing gradient-based interventions (GBIs) as a causal probing technique is to enable interventions over arbitrarily-encoded LLM representations. GBIs allow users to flexibly specify the class of representations they wish to target, expanding the scope of causal probing to arbitrarily-encoded properties. We take inspiration from Kos et al. (2018), who developed a technique to perturb latent representations using gradient-based adversarial attacks.[3] They begin by

---

[3] Notably, Tucker et al. (2021) developed a similar methodology without explicit use of such attacks (see Section 7).

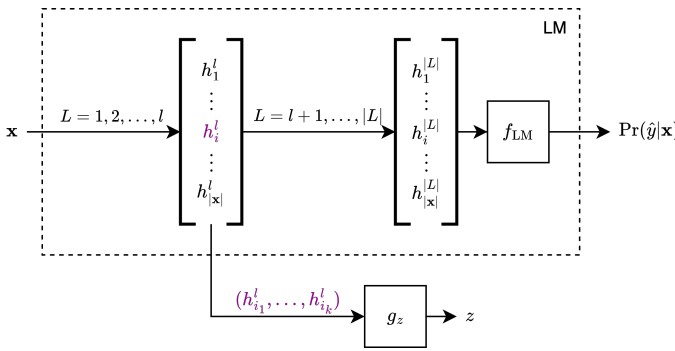

Figure 3: **Gradient-Based Interventions.** Input tokens $\mathbf{x} = (x_1, ..., x_{|\mathbf{x}|})$ are passed through layers $L = 1, ..., l$, where embedding $\mathbf{h}_i^l$ (encoding the value $Z = z$) is extracted from layer $l$ and given to $g_Z$ as input. Next, the embedding is modified by gradient-based attacks on $g_Z$ to encode the counterfactual value $Z = z'$, then fed back into subsequent layers $L = l + 1, ..., |L|$ and language modeling head $f_{\mathrm{LM}}$ to obtain the intervened predictions $M(\mathbf{x}|\operatorname{do}(Z = z'))$.

training probe $g_Z : \mathbf{h} \mapsto z$ to predict image class $z \in Z$ from latent representations $\mathbf{h} = f_{\mathrm{enc}}(\mathbf{x})$ of images $\mathbf{x}$, where $f_{\mathrm{enc}}$ is the encoder of a VAE-GAN Larsen et al. (2016) trained on an unsupervised image reconstruction task (i.e., $f_{\mathrm{dec}}(f_{\mathrm{enc}}(\mathbf{x})) = \hat{\mathbf{x}} \approx \mathbf{x}$, for decoder $f_{\mathrm{dec}}$ and reconstructed image $\hat{\mathbf{x}}$ approximating $\mathbf{x}$). Next, gradient-based attacks like FGSM Goodfellow et al. (2015) and PGD Madry et al. (2017) are performed against $g_Z$ in order to minimally manipulate $\mathbf{h}$ such that it resembles encoded representations of target image class $Z = z'$ (where $z' \neq z$, the original image class), yielding perturbed representation $\mathbf{h}'$. Finally, $\mathbf{h}$ and $\mathbf{h}'$ are each fed into the VAE decoder to reconstruct corresponding output images $\hat{\mathbf{x}}$ and $\hat{\mathbf{x}}'$ (respectively), where $\hat{\mathbf{x}}$ resembles input image class $Z = z$ and $\hat{\mathbf{x}}'$ resembles target class $Z = z'$.

We reformulate this approach in the context of causal probing as visualized in Figure 3, treating layers $L = 1, ..., l$ as the encoder and layers $L = l + 1, ..., |L|$ (composed with language modeling head $f_{\mathrm{LM}}$) as the decoder, allowing us to target representations of property $Z$ across embeddings $\mathbf{h}_i^l$ of token $x_i \in \mathbf{x}$ in layer $l$. We train $g_Z$ to predict $Z$ from a set of such $\mathbf{h}_i^l$, then attack $g_Z$ using FGSM and PGD to intervene on $\mathbf{h}_i^l$ (representing the original value $Z = z$), producing $\mathbf{h}_i^{l'}$ (representing the counterfactual value $Z = z'$). Finally, we replace $\mathbf{h}_i^l$ with $\mathbf{h}_i^{l'}$ in the LLMs' forward pass from layers $L = l + 1, ..., |L|$, simulating the intervention $\operatorname{do}(Z = z')$, and observe the impact on its word predictions $M(\mathbf{x}|\operatorname{do}(Z = z'))$.

**Benefits and Drawbacks** The key advantage of gradient-based interventions (GBIs) as a causal probing methodology is that they may be applied to any differentiable probe. For example, if we are investigating the hypothesis that $M$'s representation of $Z$ is captured by a linear subspace of representations in a given layer (see Vargas & Cotterell, 2020), then we may train a linear probe and various nonlinear probes on representations and observe whether GBIs against the linear probe have a comparable impact to those against the nonlinear probes. Alternatively, if we believe that a probe's architecture should mirror the architecture of the model it is probing (as argued by Pimentel et al., 2022), we may implement probes as such. Finally, where previous intervention methodologies for causal probing have focused on *nullifying* interventions that remove the representation of the target property $Z$ Ravfogel et al. (2020; 2022b;a); Shao et al. (2022); Belrose et al. (2024), GBIs allow one to perform targeted interventions that set LLMs' representations to counterfactual values $do(Z = z')$, effectively simulating the model's behavior under counterfactual inputs, which may be useful for predicting behaviors under various distribution shifts (see Appendix C.1). However, the benefits associated with GBIs do come with some important limitations, as we discuss in Appendix A.

## 4 APPLICATIONS OF CALM

Once we instantiate the general CALM framework with a specific probing technique such as the GBI introduced in the previous section, CALM would be "operational" and can be used in many

novel ways to both facilitate understanding of representations learned in LLMs and predict behaviors of LLMs in many application contexts, which would otherwise be impossible without such a framework. We briefly discuss some of them below for the purpose of demonstrating the generality and great potential of CALM.

### 4.1 REPRESENTATION LEARNING

The CALM framework, competence measure, and GBI methodology developed in Sections 2 and 3 are sufficiently general to be directly applied to analyze arbitrary LLMs on any language modeling task whose causal structure is already well understood (or, for tasks where this is not the case, we may apply the causal graph discovery approach described in Section 4.4), allowing us to study the impact of various model architectures, pre-training regimes, and fine-tuning strategies on the representations LLMs learn and use for arbitrary tasks of interest. For example, just using the proposed competence measure as an additional dimension of evaluation as we have done in our experiments should already enable obtaining additional insights about the behaviors of the models. As the competence measure can be expected to have better correlation with the behavior of a model than a regular task performance measure, using the competence measure or using it in addition to regular performance measures can lead to better decisions in optimizing all kinds of decisions such as model architecture and hyperparameters.

### 4.2 MULTITASK LEARNING

Are high competence scores on task $\mathcal{T}$ correlated with an LLMs' robustness to meaning-preserving transformations (see, e.g., Elazar et al., 2021a) on tasks $\mathcal{T}'$ that share several causal properties $\mathbf{Z}_c$ with task $\mathcal{T}$? Through the lens of causally invariant prediction (Peters et al., 2016; Arjovsky et al., 2019; Bühlmann, 2020), this hypothesis is likely true (however, see Rosenfeld et al. 2020 for appropriate caveats) – if so, this would make it possible to use clusters of related tasks to predict LLMs' robustness (and other behavioral patterns, such as brittleness in the face of distribution shifts introduced by spurious dependencies) between related tasks using CALM, given an appropriate experimental model. Furthermore, the ability to characterize tasks based on mutual (learned) dependency structures could be valuable in transfer learning applications such as guiding the selection of auxiliary tasks in multi-task learning (Ruder, 2017) or predicting the impact of intermediate task fine-tuning on downstream target tasks (Choshen et al., 2022).

### 4.3 TASK DEPENDENCIES

Another possible application of CALM concerns causal invariance under multi-task applications. Existing approaches in invariant representation learning generally require task-specific training Zhao et al. (2022), as the notion of invariance is inherently task-centric (i.e., the properties which are invariant predictors of output values vary by task, and different tasks may have opposite notions of which properties are causal versus environmental; see Section 2.2), so applying such approaches to train models to be causally invariant with respect to a specific downstream task $\mathcal{T}$ is expected to come at the cost of performance on other downstream tasks $\mathcal{T}'$. Therefore, considering the recent rise of open-ended, task-general LLMs Zhang et al. (2022); BigScience et al. (2022); Touvron et al. (2023a;b); Groeneveld et al. (2024), it is important to find alternative approaches for studying models' causal dependencies in a task-general setting to account for applications involving tasks with different (and perhaps contradictory) causal structures, such as CALM.

### 4.4 CAUSAL COMPETENCE GRAPH DISCOVERY

One of the key benefits of CALM is that, instead of simply measuring consistency with respect to a known, static task description $\mathcal{G}_{\mathcal{T}}$, the competence metric in Equation (2) can also be used to discover a competence graph $\mathcal{G}$ which most faithfully explains a model $M$'s behavior in a given task or context (see Section 2.3) by computing $\mathcal{C}(M|\mathcal{G})$ "in-the-loop" of existing causal graph discovery algorithms like IGSP (Yang et al., 2018). Such algorithms can be used both to suggest likely competence graphs based on interventional data collected by running CALM experiments, to recommend the experiments that would yield the most useful interventional data for the graph discovery algorithm, and to evaluate candidate graphs $\mathcal{G}$ using our competence metric, terminating the graph discovery algorithm once a competence graph $\mathcal{G}$ that offers sufficiently faithful explanations of $M$'s behavior

has been found. In this case, it is still necessary to define the set of properties $\mathbf{Z}$ being probed and the scoring function $S$ used to compare the predictions of $M$ and $\mathcal{G}$; but no knowledge of the causal dependencies (or structural functions $F : \mathbf{pa}(Z_j) \mapsto Z_j$ mapping from causal parents $\mathbf{pa}(Z_j)$ to causal dependents $Z_j$; see Bongers et al. 2021) is required.

These are only some of the possible applications enabled by the new CALM framework. One can easily imagine other possibilities, but a full discussion of all those possibilities is out the scope of this paper.

## 5 EXPERIMENTS

As the main contribution of our work is a theoretical framework, a general competence measure, and a general perturbation method, our experiments are mainly to evaluate its feasibility in studying specific models on some specific tasks to measure and understand their competence, with the hope of generating quantitative measures of the competence of LLMs for the first time.

We begin by examining BERT Devlin et al. (2019) and RoBERTa Liu et al. (2019),[4] two language models which have been extensively studied in the context of probing Rogers et al. (2020); Ravfogel et al. (2020); Liu et al. (2021); Elazar et al. (2021b); Lasri et al. (2022). Our primary goal in the following experiments is to develop and test an experimental implementation of CALM using GBIs in the context of comparatively small, well-studied models and tasks in order to validate whether CALM can explain behavioral findings of earlier work in this simplified environment. (We motivate this choice in greater detail in Appendix B.1.)

### 5.1 TASKS

Masked language models like BERT and RoBERTa are trained to predict $\Pr(x_{\texttt{[MASK]}} = w | \mathbf{x})$ for text input (token sequence) $\mathbf{x} = (x_1, x_2, ..., x_{|\mathbf{x}|})$, mask token $x_{\texttt{[MASK]}} \in \mathbf{x}$, and token vocabulary $V = \{w_1, w_2, ..., w_{|V|}\}$. As such, it is common to study them by providing them with "fill-in-the-blank" style masked prompts (e.g., "a cat is a type of $\texttt{[MASK]}$") and evaluating their accuracy in predicting the correct answer (e.g., "animal", "pet", etc.), a task known cloze prompting (Liu et al., 2023).

We use the collection of 14 lexical inference tasks included in the ConceptNet Speer et al. (2017) subset of LAMA Petroni et al. (2019), each of which are formulated as a collection cloze prompts. For example, the LAMA "IsA" task contains $\sim$2K hypernym prompts corresponding to the "IsA" ConceptNet relation (including, e.g., "A laser is a $\texttt{[MASK]}$ which creates coherent light.", where the task is to predict that the $\texttt{[MASK]}$ token should be replaced with "device", a hypernym of "laser"), with the remaining 13 LAMA ConceptNet tasks corresponding to other lexical relations such as "PartOf", "HasProperty", and "CapableOf". (See Appendix B.2 for additional details.)

Using these task datasets allow us to test how the representation of each relation is used across all other tasks. In the context of a single task $\mathcal{T}_j$, intervening on a model's representation of the task-causal relation $Z_j$ allows us to measure the extent to which its predictions are attributable to its representation of the causal property $\mathbf{Z}_c = \{Z_j\}$ (where a large impact indicates competence). On the other hand, intervening on the representations of the other 13 lexical relations $Z_k \in \mathbf{Z}_e$ allows us (in the aggregate) to measure how much the model is performing task $\mathcal{T}_j$ by leveraging representations of general, non-causal lexical information (where a large impact indicates incompetence).[5]

### 5.2 EXPERIMENTALLY MEASURING COMPETENCE

Given LLM $M$ and task $\mathcal{T}$, measuring the competence $\mathcal{C}_{\mathcal{T}}(M|\mathcal{G}_{\mathcal{T}})$ of $M$ given $\mathcal{G}_{\mathcal{T}}$ requires us to specify an experimental model $E = (\mathbf{Z}, \mathcal{G}_{\mathcal{T}}, S)$, where $\mathbf{Z}$ is a set of properties, $\mathcal{G}_{\mathcal{T}}$ is a competence graph for task $\mathcal{T}$, and $S$ is a scoring function that compares the predictions of $M$ and $\mathcal{G}_{\mathcal{T}}$. Given that each task $\mathcal{T}_i$ is defined by a single causal lexical relation $Z_i$ (i.e., $\mathbf{Z}_{c_i} = \{Z_i\}$), we model settings $\mathbf{z}$

---

[4]Specifically, $\texttt{BERT-base-uncased}$ and $\texttt{RoBERTa-base}$ Wolf et al. (2019).

[5]Note that the strictest interpretation of this formulation of competence makes the simplifying assumption that each non-causal property is equally (un)related to the target property, which is not generally true; see Appendix A.

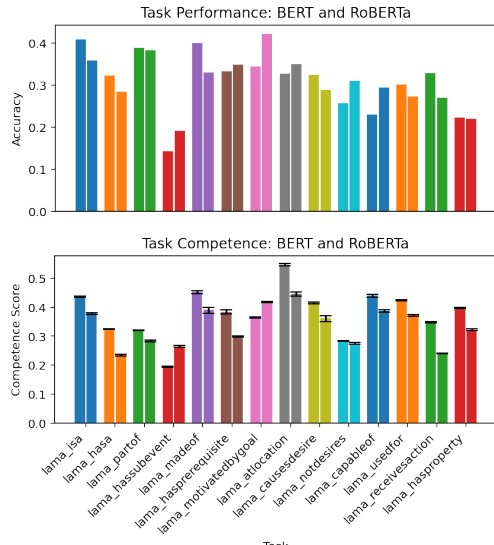

Figure 4: **Competence and Performance Results.** Performance (top) and competence (bottom) of BERT (left bars) and RoBERTa (right bars) for all tasks, using FGSM with $\epsilon = 0.1$. In the competence plot, y-values are the average competence score and error bars are the maximum and minimum competence score, as measured over 10 experimental iterations (each with a different randomly-initialized probe $g_Z$).

as a collection of values $Z_j = z_j$ taken by each property $Z_j$ in the context of a specific task instance $(\mathbf{x}, \mathbf{y}) \sim \mathcal{T}_i$, where $Z_j = 1$ if $i = j$ (i.e., where the property $Z_j$ is the causal property for the task $\mathcal{T}_i$) or $Z_j = 0$ otherwise. That is, for each instance $(\mathbf{x}, \mathbf{y}) \sim \mathcal{T}_i$, the corresponding setting $\mathbf{z}$ is a one-hot vector whose $i$-th element $\mathbf{z}_i = 1$. We may specify $\mathcal{G}_{\mathcal{T}_i}$ in a similar manner: for task $\mathcal{T}_i \sim P(\mathcal{X}, \mathcal{Y})$, outputs $\mathbf{y} \in \mathcal{Y}$ are causally dependent on the property $Z_i$, and invariant to other concepts $Z_j, j \neq i$., meaning that the only direct parent node of $\mathbf{y}$ in $\mathcal{G}_{\mathcal{T}_i}$ is $Z_i$. Finally, as we are dealing with masked language models whose output space $\mathcal{Y}$ for each task consists only of single tokens in $M$'s vocabulary $V_M$, our experimental model can define the scoring function $S$ as the overlap $\text{overlap}(\mathbf{y}_i, \mathbf{y}_j)$ for top-$k$ token predictions $\mathbf{y}_i = \{y_1, ..., y_k\} \subset V_M$, where $\text{overlap}(\cdot, \cdot)$ is the size of the intersection of each set of predictions divided by the total number of predictions $\text{overlap}(\mathbf{y}_i, \mathbf{y}_j) = \frac{|\mathbf{y}_i \cap \mathbf{y}_j|}{k}$. (See Appendix C.2 for additional details on how we compute competence in each experiment.)

## 5.3 PROBES

We implement probes $g_Z$ as a 2-layer MLP over each language model's final hidden layer, and train the probe on the task of classifying whether there is a particular relation $Z$ between a final-layer `[MASK]` token in the context of a cloze prompt and the final-layer object token from the "unmasked" version of the same prompt. All reported figures are the average of 10 runs of our experiment, using different randomly-initialized $g_Z$ each time. (See Appendix B.3 for further details.)

## 5.4 INTERVENTIONS

We implement GBIs against $g_Z$ using two gradient attack strategies, FGSM Goodfellow et al. (2015) and PGD Madry et al. (2017). We bound the magnitude of each intervention as follows: where $h$ is the input to $g_Z$ and $h'$ is the intervened representation following a GBI, $||h - h'||_\infty \leq \epsilon$. For all experiments reported in our main paper, we use FGSM with $\epsilon = 0.1$. (See Appendix B.4 for more details and PGD results.)

## 6 RESULTS

In Figure 4, we visualize the performance and competence of BERT and RoBERTa across the test set of each LAMA ConceptNet task. Performance is measured using $(0, 1)$-accuracy, competence is measured using the experimental competence metric in Equation (3), and both metrics are averaged across the top-$k$ predictions of each model for $k \in [1, 10]$. Specifically, for accuracy, we compute

$$\frac{1}{n} \sum_{k=1}^{n} \mathbb{1}[y \in \underset{\hat{y}}{\text{top-}k} \Pr(\hat{y}|\mathbf{x})]$$

for ground truth $(\mathbf{x}, y)$ and $n = 10$; and for competence, we compute

$$\frac{1}{n} \sum_{k=1}^{n} \mathcal{C}_{\mathcal{T}}(M|\mathcal{G}_{\mathcal{T}})$$

To account for stochasticity in initializing and training probes $g_Z$, scores are also averaged over 10 randomized experiments for each target task where the probe is randomly re-initialized each time (resulting in different GBIs).

### 6.1 ANALYSIS

**Performance** While their accuracies on individual tasks vary, BERT and RoBERTa have quite similar aggregate performance: BERT outperforms RoBERTa on just over half (8/14) of the tasks, achieving essentially equivalent performance when averaged across all tasks (0.3099 versus 0.3094).

**Competence** Given our experimental model $E$ with $m = 14$ tasks, consider a random baseline language model $R$ whose predictions always change in response to each intervention, making equal use of all properties in each task. $R$ would yield a competence score of $\mathcal{C}(R|\mathcal{G}_{\mathcal{T}}) = \frac{1}{m} \approx 0.0714$ for each task. Both BERT and RoBERTa score above this threshold for all tasks, meaning that their competence is consistently greater than that of a model ($R$) that does not distinguish between causal and environmental properties. However, RoBERTa is consistently less competent than BERT (on 12/14 tasks), and also has lower competence scores averaged across all tasks (0.381 vs. 0.334).

We also observe that, for the two tasks (HasSubevent and MotivatedByGoal) where RoBERTa is more competent than BERT, it also achieves substantially higher performance. Specifically, relative performance and competence are correlated: the Spearman's Rank correlation coefficient between the average difference in accuracy and average difference in performance is a moderately strong positive correlation $\rho = 0.508$ with significance $p = 0.064$.

### 6.2 DISCUSSION

*A priori*, we might expect an LLM with nontrivial performance to also exhibit greater competence than a random baseline like $R$; but this is not necessarily the case, given that it is common for deep learning models to achieve remarkable performance by exploiting spurious correlations inherent in a given task dataset McCoy et al. (2019); Geirhos et al. (2020); Feder et al. (2022). Thus, the finding that BERT and RoBERTa's performance on each task is supported by an intermediate level of competence on the part of both models is meaningful: for each task, their behavior is generally more attributable to their representations of causally-invariant properties than to spurious lexical associations, and this competence varies substantially between tasks.

**Explananda** Prior work has shown that BERT and RoBERTa have widely varying performance in response to lexical inference tasks, depending on the specific manner in which they are prompted (Hanna & Mareček, 2021; Ravichander et al., 2020; Ettinger, 2020; Elazar et al., 2021a; see Section 7). One possible explanation for this phenomenon is that these models may not consistently utilize a representation of the task-causal lexical relations (i.e., they are not highly competent for these tasks), instead relying (at least in part) on spurious lexical associations learned from its training data. Previously, it has not been possible to empirically assess this hypothesis; but using CALM, it is possible to provide direct evidence in its favor, as we find that both models possesses (only) an intermediate degree of competence for lexical inference prompt tasks.

## 7 RELATED WORK

**Hypernym Prompting**   The performance of BERT-like models on lexical inference tasks such as hypernym prediction is known to be highly variable under small changes to prompts Hanna & Mareček (2021); Ravichander et al. (2020); Ettinger (2020); Elazar et al. (2021a). Our findings offer one possible explanation for such brittle performance: BERT and RoBERTa's partial competence in hypernym prediction indicates that it should be possible to prompt these models in a way that will yield high performance, but that its reliance on spurious lexical associations may lead it to fail when these correlations are broken – e.g., by substituting singular terms for plurals Ravichander et al. (2020) or paraphrasing a prompt Elazar et al. (2021a).

**Causal Probing**   Most related to our work is amnesic probing Elazar et al. (2021b), which we discuss at length in Section 2.4. Lasri et al. (2022) applied amnesic probing to study the use of grammatical number representations in performing an English verb conjugation prompt task. As this experiment involves intervening on the representation of a property which is causal with respect to the prompt task, it may be understood as an informal instantiation of CALM (albeit without considering environmental properties or measuring competence).

**Gradient-based Interventions**   Tucker et al. (2021) developed a similar approach to our GBI causal probing methodology (as outlined in Section 2.4) without explicit use of gradient-based adversarial attacks. Their methodology is equivalent to performing a targeted, unconstrained attack using standard gradient descent.[6] In such attacks, it is standard practice to constrain the magnitude of resulting perturbations Goodfellow et al. (2015); Madry et al. (2017); Kos et al. (2018), which we do here in order to minimize the effect of "collateral damage" done by such attacks (see Section 5.4 and Appendix B.4); so failing to impose such constraints may result in indiscriminate damage to representations.

**Unsupervised Probing**   Instead of training supervised probes to predict a pre-determined property of interest (as we do here), an alternative approach is to train *unsupervised* probes such as Sparse Auto-Encoders (SAEs; Subramanian et al., 2018; Yun et al., 2021; Cunningham et al., 2023) to automatically learn an overcomplete basis of features that are useful for sparsely representing embeddings, which can also be used to control models' use of these learned features Bricken et al. (2023); Templeton et al. (2024). However, as SAEs are unsupervised probes, they yield feature vectors that are not inherently interpretable and must be retroactively interpreted, meaning that the task of creating a supervised probe training dataset (as required for conventional causal probing) is substituted for the task of interpreting learned features Davies & Khakzar (2024). However, given features that can be reliably interpreted as representing task-causal or -environmental features, it is also possible to implement CALM using unsupervised probes like SAEs.

## 8 CONCLUSION

In this work, we introduced CALM, a general analysis framework that enables the study of LLMs' linguistic competence using causal probing, including the first quantitative measure of linguistic competence. We developed the gradient-based intervention (GBI) methodology, a novel approach to causal probing that can target a far greater range of representations than previous techniques, expanding the scope of causal probing to new questions in LLM interpretability and analysis. We discussed multiple new applications of CALM in analyzing and understanding the learned representations of LLMs as well as predicting their behaviors. Finally, we carried out a preliminary case study of CALM using GBIs, analyzing BERT and RoBERTa's competence across a collection of lexical inference tasks, finding that even a simple experimental model is sufficient to explain and predict their behavior across a variety of lexical inference tasks. These results demonstrated the great potential of CALM in studying representations and behaviors of LLMs in novel ways that we could not do today.

---

[6]I.e., they continue running gradient updates until the targeted probe loss saturates, irrespective of resulting perturbation magnitude.

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

## A  LIMITATIONS

**Gradient-Based Interventions**  While GBIs are applicable to a more general range of model representations than other interventions (see Section 3), this generality comes with a lack of constraints on probes ($g_Z$); and as a result, GBIs cannot provide the strong theoretical constraints on collateral damage as can methods like, e.g., INLP Ravfogel et al. (2020), which provably preserves distances between embeddings as well as possible while completely removing the linear representation of the target property. To minimize collateral damage to representations, the magnitude of perturbations should be modulated via constraints on gradient attacks against $g_Z$ (see Section 5.4) and experimentally validated to control the damage done to representations (see Appendix B.4). Thus, in cases where the structure of representations is believed to satisfy strong assumptions (e.g., being restricted to a linear subspace; Vargas & Cotterell, 2020) or strong upper bounds on collateral damage are required, CALM interventions can be implemented with methods like INLP rather than GBIs.[7]

---

[7]It may also be possible to control for collateral damage by developing GBI strategies that offer more principled protection against damage to non-targeted properties, such as adding a loss term to penalize dam-

**Tasks** In our experiments, we modeled the 14 LAMA ConceptNet tasks as representing fully independent properties, which is not necessarily true – e.g., knowing that a tree is made of bark or contains leaves tells us something about whether it is a type of plant. However, in the aggregate (with impacts summed across 14 widely-varying lexical relation types in computing the final competence score for each task; see Appendix C.2), it may nonetheless be appropriate to treat the relations which are not causal with respect to a given task as collectively capturing spurious lexical associations.

## B EXPERIMENTAL DETAILS

### B.1 SIMPLIFIED ENVIRONMENT

As noted in Section 5, our primary goal in our experiments is to validate CALM by testing it in a simplified experimental setting consisting of comparatively small, well-studied models and tasks. As such, we need models that are *just complex enough* for CALM to be applicable (i.e., neural language models that are capable of performing the tasks we consider at a nontrivial level of performance), making BERT and RoBERTa ideal candidates; and in future work plan to scale CALM to more complex contexts covering larger, more powerful models as they perform more difficult tasks (see **??**). This is a common setting in the context of substantial recent interpretability work: first, a theoretical framework is developed for interpreting an internal representation or mechanism and initially tested in the context of "toy" models or tasks Elhage et al. (2021); Olsson et al. (2022); Zhong et al. (2023); Geiger et al. (2023), and subsequent work scales these frameworks to the context of larger models "in the wild" Wang et al. (2023b); Conmy et al. (2023); Wu et al. (2023). We anticipate that all of our major contributions (the CALM framework, competence metric, and GBI causal probing method) will in principle be scalable to much larger, more recent LLMs (e.g., Zhang et al. 2022; BigScience et al. 2022; Touvron et al. 2023a;b; Groeneveld et al. 2024, etc.), and predict that the main challenge will be in finding an appropriate probing architecture (see Pimentel et al. 2022).

### B.2 TASKS

The full set of LAMA ConceptNet tasks is as follows: IsA, HasA, PartOf, HasSubEvent, MadeOf, HasPrerequisite, MotivatedByGoal, AtLocation, CausesDesire, NotDesires, CapableOf, UsedFor, ReceivesAction, and HasProperty. We split each task dataset into train, validation, and test sets with a random $80\%/10\%/10\%$ split. Train and validation instances are fed to each model to produce embeddings used to train $g_Z$ and select hyperparameters, respectively; and test instances are used to measure LLMs' competence with respect to each task by observing how predictions change under various interventions. In all experiments, we restrict each model $M$'s output space for each task $\mathcal{T}$ to the subset of vocabulary $V_M$ that occurs as a ground-truth answer $y^*$ for at least one instance $(\mathbf{x}, y^*) \sim \mathcal{T}$ in the respective task dataset. This lowers the probability of false negatives in evaluation (e.g., penalizing the model for predicting $\hat{y} =$ "mammal" for "a dog is a type of $y$" instead of $y^* =$ "animal").

### B.3 PROBES

We use BERT's final layer $L$ to encode $h_i^l$ embeddings for each such example, where $i$ is the index of the [MASK] token or target word in the input prompt $x_i$. To encode the [MASK] token, we issue BERT masked prompts (as discussed above) to extract $h_{\text{[MASK]}}$, then repeat with the [MASK] token filled-in with the target word to encode it as $h_+$ (e.g., "device" in "A laser is a device which creates coherent light."), and concatenate matching embeddings $h = (h_{\text{[MASK]}}; h_+)$ to produce positive ($y = 1$) training instances. We also construct one negative ($y = 0$) instance, $h = (h_{\text{[MASK]}}; h_-)$, for each $h_{\text{[MASK]}}$ by sampling an incorrect target word $x_i$ corresponding to an answer to a random prompt from the same task, feeding it into the cloze prompt in the place of the correct answer, and obtaining BERT's contextualized final-layer embedding of this token ($h_-$). Finally, we train $g_Z$ on the set of all such $(h, y)$.

---

age to non-targeted probes or leveraging interval bound propagation Gowal et al. (2019) to place intervened embeddings inside the adversarial polytope for non-targeted properties. We leave such possibilities to future work.

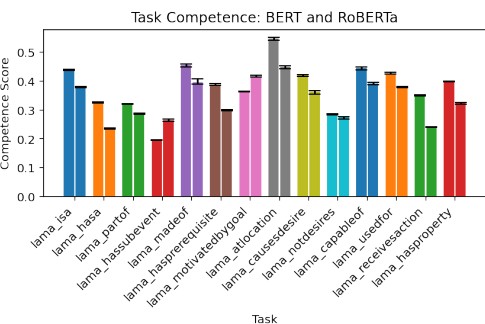

Figure 5: Competence of BERT (left bars) and RoBERTa (right bars) for all tasks, using PGD with $\epsilon = 0.1$. Y-values are the average competence score and error bars are the maximum and minimum competence score, as measured over 10 experimental iterations (each with a different randomly-initialized probe $g_Z$).

We implement $g_Z$ as a multi-layer perceptron with 2 hidden layers, each with a width of 768 (which is one half the concatenated input dimension of 1536), using ReLU activations and dropout with $p = 0.1$, training it for 32 epochs using Binary Cross Entropy with Logits Loss[8] and the Adam optimizer, saving the model from the epoch with the highest validation-set accuracy for use in all experiments.

For all competence results reported in Section 6, we run the same experiment 10 times – each with a different random initialization of $g_Z$ and shuffled training data – and report each figure as the average among all 10 runs.

### B.4 INTERVENTIONS

For instance $(h, y)$, classifier $g_Z$, loss function $\mathcal{L}$, and $L_\infty$-bound $\epsilon \in \{0.01, 0.03, 0.1, 0.3\}$[9], each intervention (gradient attack) $g_z$ may be used to produce perturbed representations $h' = g_z(h, y, f_{\text{cls}}, \mathcal{L}, \epsilon)$ where $||h - h'||_\infty \leq \epsilon$. In particular, given $h = (h_{\texttt{[MASK]}}; h_\pm) \in \mathbb{R}^{2d}$, let $h'_{\texttt{[MASK]}}$ be the first $d$ dimensions of $h'$ (which also satisfies the $L_\infty$-bound with respect to $h_{\texttt{[MASK]}}$, $||h_{\texttt{[MASK]}} - h'_{\texttt{[MASK]}}||_\infty \leq \epsilon$). To measure BERT's use of internal representations of $Z$ on each prompt task, we evaluate its performance when perturbed $h'_{\texttt{[MASK]}}$ is used to compute masked-word predictions, compared to unperturbed $h_{\texttt{[MASK]}}$.

Our intent in intervening only on the final-layer mask embedding $h_{\texttt{[MASK]}}$ in our experiments is that, in the final layer of a masked language model such as BERT or RoBERTa, the only embedding which is used to compute masked-word probabilities is that of the $\texttt{[MASK]}$ token. Thus, any representation of the property that is *used* by the model in its final layer must be a part of its representation of the $\texttt{[MASK]}$ token, preventing "recoverability" phenomena such as those observed by Elazar et al. (2021b).

**FGSM** We implement Fast Gradient Sign Method (FGSM; Goodfellow et al., 2015) interventions as

$$h' = h + \epsilon \cdot \text{sgn}(\nabla_h \mathcal{L}(f_{\text{cls}}, x, y))$$

**PGD** We implement Projected Gradient Descent (PGD; Bubeck et al., 2015; Madry et al., 2017) interventions as $h' = h^T$ where

$$h^{t+1} = \Pi_{N(h)}\big(h^t + \alpha \cdot \text{sgn}(\nabla_h \mathcal{L}(f_{\text{cls}}, x, y))\big)$$

---

[8]https://pytorch.org/docs/stable/generated/torch.nn.BCEWithLogitsLoss.html

[9]All reported results use $\epsilon = 0.1$, as greater $\epsilon$ resulted in unacceptably high "collateral damage" across target tasks (e.g., even random perturbations of magnitude $\epsilon = 0.3$ do considerable damage), and lesser values meant that predictions changed on target tasks consisted of only a few test instances.

for iterations $t = 0, 1, ..., T$, projection operator $\Pi$, and $L_\infty$-neighborhood $N(h) = \{h' : ||h - h'|| \leq \epsilon\}$. This method also introduces two hyperparameters, the number of PGD iterations $T$ and step size $\alpha$. We use hyperparameter grid search over $\alpha \in \{0.001, 0.003, 0.01, 0.03\}$ and $T \in \{20, 40, 60, 80, 100\}$, finding that setting $\alpha = \frac{\epsilon}{10}$ and $T = 40$ produces the most consistent impact on $g_Z$ accuracy across all tasks; so we use these values for the results visualized in Figure 5.

### B.5 COMPUTE BUDGET

BERT-base-uncased has 110 million parameters, and RoBERTa-base has 125M parameters. As our goal is to study the internal representation and use of linguistic properties in existing pre-trained models, and we are not directly concerned with training or fine-tuning such models, we use these models only for inference (including encoding text inputs, using embeddings to train probes, and feeding intervened embeddings back into the language models). The only models we trained were probes $g_Z$, which each had 1.77M parameters.

Each experimental iteration (including encoding text inputs, training probes on all 14 tasks, and performing all GBIs) for either BERT or RoBERTa took less than one hour on a single NVIDIA GeForce GTX 1080 GPU, meaning that running all 10 iterations across both language models took less than 20 hours on a single GPU. Each iteration, probe, and GBI can easily be parallelized across GPUs: in our case, running all iterations across both models took less than 3 hours total across 8 GTX 1080 GPUs.

## C COMPETENCE METRIC

### C.1 COMPARISON WITH IIA

As noted in Section 2.3, the $\mathcal{C}_\mathcal{T}(M|\mathcal{G}_\mathcal{T})$ metric defined in Equation (2) is an adaptation of the Interchange Intervention Accuracy (IIA) metric (Geiger et al., 2022; 2023), which evaluates the faithfulness of a causal abstraction like $\mathcal{G}_\mathcal{T}$ as a (potential) explanation of the behavior of a "black box" system like $M$. In our case, this is equivalent to evaluating the competence of $M$ on task $\mathcal{T}$, provided that $\mathcal{G}_\mathcal{T}$ is the appropriate SCM for $\mathcal{T}$, as an LLM is competent only to the extent that its behavior is determined by a causally invariant representation of the task.[10] IIA requires performing *interchange interventions* $M(\mathbf{x}_i | \operatorname{do}(\mathbf{z}_i))$, where the part of $M$'s intermediate representation of input $\mathbf{x}_i$ hypothesized to encode latent variables $\mathbf{Z}$ (taking the values $\mathbf{z}_i$ when provided input $\mathbf{x}_i$) is replaced with that of $\mathbf{x}_j$ (which, in the ideal case, causes $M$'s representation to encode the values $\mathbf{z}_j$ instead of $\mathbf{z}_i$), and compute the accuracy of $\mathcal{G}_\mathcal{T}(\mathbf{x}_i | \operatorname{do}(\mathbf{z}_j))$ in predicting $M$'s behavior under these interventions. Thus, given access to high-quality interchange interventions over $M$, IIA measures the extent to which $\mathcal{G}_\mathcal{T}$ correctly models $M$'s behavior under counterfactuals, and thus its faithfulness as a causal abstraction of $M$.

To adapt IIA to the context of causal probing and define $\mathcal{C}_\mathcal{T}(M|\mathcal{G}_\mathcal{T})$, we replace instance-level interchange interventions with concept-level interventions: instead of swapping $M$'s representation of variables $\mathbf{Z}$ given input $\mathbf{x}_i$ with that of $\mathbf{x}_j$, we intervene on representations at the level of arbitrary concept settings $\mathbf{z}$ that need not correspond to previously sampled $\mathbf{x}$, allowing us to simulate the behavior of $M$ under previously-unseen distribution shifts (i.e., settings $\mathbf{z}$ representing previously-unseen combinations of property values) and therefore make broader predictions about $M$'s consistency with a given causal model $\mathcal{G}_\mathcal{T}$ under such conditions. As one of the key desiderata in studying LLM competence is to predict behavior under distribution shifts where spurious correlations are broken, $\mathcal{C}_\mathcal{T}$ is more appropriate than IIA in this setting. However, it also introduces an additional challenge: where interchange interventions only require localizing candidate representations – as counterfactual representations are obtained merely by "plugging in" values from a different input – computing $\mathcal{C}_\mathcal{T}$ instead requires one to both localize representations and directly intervene on them to change the encoded value. Previous causal probing intervention strategies (e.g., Ravfogel et al., 2020; 2022b) have generally performed interventions by *neutralizing* concept representations, not modifying them to encode specific counterfactual values; so in order to carry out our study, it is also

---

[10]For many tasks, there is more than one valid $\mathcal{G}_\mathcal{T}$ (see, e.g., the "price tagging game" constructed by Wu et al. (2023)). In such cases, $\mathcal{C}_\mathcal{T}(M|\mathcal{G}_\mathcal{T})$ should be computed with respect to each valid $\mathcal{G}_\mathcal{T}$ and the highest result should be selected, as conforming to any such $\mathcal{G}_\mathcal{T}$ carries the same implications.

$$\mathcal{C}_{\mathcal{T}}(M|\mathcal{G}_{\mathcal{T}}) \approx \frac{1}{n \cdot m} \sum_{i=1}^{n} \sum_{j=1}^{m} \text{overlap}\left(M\big(\mathbf{x}_i|\,\text{do}(Z_j = 0)\big), \mathcal{G}_{\mathcal{T}}\big(\mathbf{x}_i|\,\text{do}(Z_j = 0)\big)\right) \tag{3}$$

necessary to develop a novel approach to perform such interventions. We develop a solution to this problem, gradient-based interventions (GBIs), in Section 3.

## C.2 EXPERIMENTAL COMPETENCE METRIC

To compute the expectation in Equation (2) for test set $\{\mathbf{x}_i, \mathbf{y}_i, \mathbf{z}_i\}_{i=1}^{n} \sim \mathcal{T} \times \mathbf{Z}$, we sum the competence score over all samples $\mathbf{x}_i$ and perform one intervention $\text{do}(Z_j = 0)$ corresponding to each concept $Z_j \in \mathbf{Z}$.[11] As our goal is to measure the extent to which $M$'s behavior is attributable to an underlying representation of the causal property $Z_c$ or environmental property $Z \in \mathbf{Z}_e$, our experimental model defines $\mathcal{G}_{\mathcal{T}}$'s predictions with reference to $M$'s original predictions $M(\mathbf{x}_i) = \hat{\mathbf{y}}_i$, according to the following principle: if $M$ is competent, then its prediction $M(\mathbf{x}_i) = \hat{\mathbf{y}}_i$ is wholly attributable to its representation of causal property $Z_c$, so its predictions $M(\mathbf{x}_i|\,\text{do}(Z_c)) = \hat{\mathbf{y}}_i'$ will not overlap with its original predictions $\hat{\mathbf{y}}_i$ (i.e., $\text{overlap}(\hat{\mathbf{y}}_i, \hat{\mathbf{y}}_i') = 0$); and conversely, a competent $M$ will make the *same* predictions $M(\mathbf{x}_i|\,\text{do}(Z_j)) = \hat{\mathbf{y}}_i''$ for any $Z_j \in \mathbf{Z}_e$, because its prediction is not caused by its representation of these environmental properties (i.e., $\text{overlap}(\hat{\mathbf{y}}_i, \hat{\mathbf{y}}_i'') = 1$). Motivated by this reasoning, our experimental model defines $\mathcal{G}_{\mathcal{T}}(\mathbf{x}_i|\,\text{do}(Z_j = 0)) = M(\mathbf{x}_i)$ for environmental $Z_j \in \mathbf{Z}_e$; and for causal property $Z_c$, defines $\mathcal{G}_{\mathcal{T}}(\mathbf{x}_i|\,\text{do}(Z_c = 0)) = \{y' \in V_M : y' \notin M(\mathbf{x}_i)\}$ (i.e., the set of all tokens $y'$ in $M$'s vocabulary that were not in its original prediction $M(\mathbf{x}_i)$). Thus, under experimental model $E$, we approximate $\mathcal{C}_{\mathcal{T}}(M|\mathcal{G}_{\mathcal{T}})$ by computing Equation (3).

Notably, our experimental model $E$ only accounts for the relationship between $M$'s intervened and non-intervened predictions, independently of ground truth labels – instead, what is being measured is $M$'s consistency under meaning-preserving interventions $\text{do}(Z_{j'})$ and its mutability under meaning-altering interventions $\text{do}(Z_j)$. However, as we find in Section 6.1, the resulting competence metric $\mathcal{C}_{\mathcal{T}}(M|\mathcal{G}_{\mathcal{T}})$ is nonetheless useful for predicting $M$'s accuracy.

---

[11]Note that this intervention changes the prediction $\mathcal{G}_{\mathcal{T}}(\mathbf{x}_i) \neq \mathcal{G}_{\mathcal{T}}(\mathbf{x}_i|\,\text{do}(Z_j = 0))$ if and only if $(\mathbf{x}_i, \mathbf{y}_i) \in \mathcal{T}_j$ – i.e., where the corresponding $(\mathbf{z}_i)_j = 1$ – otherwise, $(\mathbf{z}_i)_j$ is already 0, so the intervention has no effect. Thus, as $\mathcal{C}_{\mathcal{T}}(M|\mathcal{G}_{\mathcal{T}})$ measures $M$'s consistency with $\mathcal{G}_{\mathcal{T}}$, then to the extent that $M$ is competent, its prediction should change under all and only the same interventions as $\mathcal{G}_{\mathcal{T}}$.

