# OpenReview forum: "Competence-Based Analysis of Language Models"
_ICLR.cc/2025/Conference — ICLR 2025 Conference Withdrawn Submission_

### Official Review · Reviewer_emzV · 2024-11-01

**Soundness:** 2
**Presentation:** 2
**Contribution:** 1
**Rating:** 3
**Confidence:** 3

**Summary:**

The paper is intended to shed light on the behaviour of LMs. The authors propose 'competence-based analysis of language models' (CALM):
- Firstly giving a causal formulation of the linguistic notion of competence in the context of LMs (roughly, invariance of the output to perturbation of irrelevant lexical features)
- Secondly giving a framework for testing this notion of competence using causal probing and preexisting gradient-based methods
- Finally giving some basic demonstrations of the framework being applied to BERT and RoBERTA on a few different tasks

**Strengths:**

- The introduction of a formal notion of competence in terms of causal models is interesting, and the application of insights from linguistics seem to be novel and original
- In general, the mathematical framework seems sound and well-presented
- The goal of the paper seems important and valuable: I am excited by the prospect of work that helps us to understand and predict the behaviour of models

**Weaknesses:**

The paper's main contribution is a theoretical framework, and I think they do not provide a particularly compelling case that the framework actually shows meaningful results. For instance, in the abstract they say "CALM can be used to explain and predict behaviors across these tasks", but as far as I can tell, what they show empirically is only the competence metric they propose is reasonably correlated with accuracy across a handful of task/model combinations. They suggest many potential contexts where the framework could be applied, but it is not clear to me what the framework would actually show.

The paper would be much stronger if the authors spelled out far more clearly what kinds of behaviours CALM might predict or explain, and then demonstrated that it does in fact predict or explain them, especially in contexts where this would be useful and significant - for example, if the paper gave concrete examples of how CALM could predict a model's performance on out-of-distribution data, or explained unexpected failures on certain linguistic phenomena.

**Nitpicks (not affecting the score):**
- Use \citep for parenthetical citations
- Define GBI when it's first used
- Define FGSM and PGD, and give some indication of how they work / how they are different
- The paper is missing a lot of articles, eg/ 158 "study *a* model M's", 159 "performing *a* task T", 254 "training *a* probe", and there are a few other missing prepositions: I recommend checking over the text carefully for these

**Questions:**

**Q1**
> "Once we instantiate the general CALM framework with a specific probing technique such as the GBI introduced in the previous section, CALM would be “operational” and can be used in many novel ways to both facilitate understanding of representations learned in LLMs and predict behaviors of LLMs in many application contexts, which would otherwise be impossible without such a framework."

This seems like a very strong claim. Could you give an example of a behaviour that could be predicted with CALM that would otherwise be impossible to predict without such a framework? Ideally a specific hypothetical scenario.

**Q2**
> "For example, just using the proposed competence measure as an additional dimension of evaluation as we have done in our experiments should already enable obtaining additional insights about the behaviors of the models. As the competence measure can be expected to have better correlation with the behavior of a model than a regular task performance measure, using the competence measure or using it in addition to regular performance measures can lead to better decisions in optimizing all kinds of decisions such as model architecture and hyperparameters."

Can you expand on what this means, and perhaps give a concrete example?

**Q3**
> "On the other hand, intervening on the representations of the other 13 lexical relations Zk ∈ Ze allows us (in the aggregate) to measure how much the model is performing task Tj by leveraging representations of general, non-causal lexical information (where a large impact indicates incompetence)."

Could you expand on what you mean by leveraging representations of non-causal lexical information? How do you distinguish this outcome from your method simply removing/replacing more information than intended? Is there a reason that the gradient-based approach you give should only perturb the intended property?

**Q4**

Line 412 - why do you use intersection in top-k rather than some kind of probability measure?

**Q5**

Eq 1: Is there a reason you use both Pr and P?

---



My review is based on the impression that this framework, applied in its current form, is unlikely to yield novel or significant insights about LM behaviour. I would be open to revising my score if the authors can address this concern. I'm particularly interested in precise situations where CALM could explain otherwise unexpected behaviours.

---

### Official Review · Reviewer_Dqxi · 2024-11-03

**Soundness:** 3
**Presentation:** 2
**Contribution:** 2
**Rating:** 3
**Confidence:** 4

**Summary:**

The paper introduces an evaluation framework, which seeks to understand language models (LMs) by examining their internal representations of linguistic properties. It uses causal probing to measure the alignment of an LM’s representations with human-understandable causal structures of specific tasks.

**Strengths:**

The strength of the paper is that it uses an evaluation approach that combines causal probing with Gradient-Based Interventions (a concept similar to adversarial perturbations). The introduction of GBIs allows for a broader and more precise range of probing interventions, addressing some possible previous limitations in probing techniques.

**Weaknesses:**

Overall the novelty of the paper is limited as it uses an existing dataset (ConceptNet), and the established method of causal probing. Could you clearly articulate what aspects of your approach are novel beyond the gradient-based intervention strategy for causal probing? It seems to me that your 5 contributions are summarized by this strategy and its application.

Framework: (1) The title seems somewhat misleading; this work is less a "Competence-Based Analysis of Language Models" and more an evaluation approach focused on an engineered relational database (ConceptNet). Moreover, it’s unclear whether this can be accurately described as a measure of "linguistic competence," given that it examines only specific semantic relations. Could the authors clarify what constitutes the "analysis" aspect of the framework? Is this not more accurately described as an evaluation framework? (2) From the title and abstract the reader expects some interpretability or causal analysis to better understand models and their behaviors. There is a need to clarify what are the interpretable aspects of this framework and how it leads to interpretable insights about model behavior. (3) Additionally, I would like to see some motivation regarding the usefulness of this framework. Maybe comparing the CALM framework with other existing causal probing frameworks could indicate its importance. (4) The study’s focus on lexical inference tasks may limit the generalizability of CALM to other types of language tasks. Could you provide some motivation as to why you focused on lexicosemantic tasks?

Experiments: (1) There is no evidence that the framework is scalable to more complex, bigger, auto-regressive models. Could the authors motivate why no auto-regressive model architectures were used? (2) The framework is tested in a simplified experimental setting consisting of comparatively small, well-studied models and tasks. (3) How is this framework useful or applicable in the recent research landscape of LLMs? (4) The authors note that model "competence varies substantially between tasks," a pattern likely to persist as this framework is expanded to additional tasks and models. Previous studies on lexical inference tasks (Hanna and Marecek, 2021; Ravichander et al., 2020; Ettinger, 2020; Elazar et al., 2021a) have also highlighted such inconsistencies in model performance. Given these known variations, what novel evaluative insights does the CALM framework provide?

**Questions:**

I would recommend making more clear the concept of *environmental properties* and the difference between casual properties. As the paper is now you only mention environmental properties 8 times without further explanation or examples.

 In the abstract, you mention investigating competence by "intervening on models’ internal representations of different linguistic properties." I suggest replacing the broad references to "linguistic properties" and " linguistic competence" with "lexicosemantic properties" and "lexicosemantic competence." Since your work specifically addresses a formal aspect of linguistic competence and not functional linguistic competence, the term "linguistic competence" may come across as overly general and potentially overstates the study's scope.

If this is a competence framework I would suggest validating the framework’s effectiveness across real-world applications to strengthen its practical relevance beyond controlled experiments. How well does CALM generalize to tasks outside of lexical inference?

Figure 4's different y-axis makes it hard to compare accuracy vs competence scores (minor but I would remove the lama_ in the x labels. It is mentioned throughout the paper that the categories are a set of LAMA ConceptNet tasks, you could mention it in the Figure caption).

Spelling:
355-356: each of which are formulated as a collection cloze prompts -> each of which **is** formulated as a collection **of** cloze prompts
361-362: Using these task datasets allow us to test how the representation of each relation is used across all other tasks. -> Using these task datasets allows us to test
485: as we find that both models **possess** (only) an intermediate degree of competence for lexical inference prompt tasks.
There are some “?” (missing citations) throughout the paper.

---

### Official Review · Reviewer_g3uh · 2024-11-03

**Soundness:** 2
**Presentation:** 2
**Contribution:** 2
**Rating:** 3
**Confidence:** 4

**Summary:**

This paper seeks to develop a framework for assessing the relative competence of LLMs at specific linguistic phenomena. In essence, this means assessing whether LLMs conform to simple causal models of how linguistic items depend on other linguistic items. The experiments focus on mostly lexical relations, and the models assessed are BERT and RoBERTa. Overall, these models are only moderately successful at the lexical tasks, and, by the paper's methods, only moderately competent as well.

**Strengths:**

It is great to see creative work that is seeking to push beyond simple behavioral assessments to more deeply understand how models represent linguistic data. This seems critical to the project of interpretability, which itself seems like an important building block for foundational work on robustness and out-of-domain generalization.

I also appreciate that the authors have defined some creative causal probing techniques.

**Weaknesses:**

1. Really a lot of relevant literature is not cited. The paper makes very few connections to causal interpretability research and very few connections to work on targeted analysis of specific linguistic phenomena.

    * For an overview of interpretability methods in this space, I recommend [Mueller et al. 2024](https://arxiv.org/abs/2408.01416) -- this article will help fill in gaps in the current paper's connections to prior wrok.

    * For linguistic analyses, the major benchmarks are SyntaxGym and BLiMP, each of which now have variants in other languages. The most closely related paper to the current one is [Arora et al. 2024](https://aclanthology.org/2024.acl-long.785/) (CausalGym). That paper extends SyntaxGym to support supervised interpretability, and it evaluates lots of different methods on the SyntaxGym/CausalGym.

4. The linguistic investigations in this paper are a good deal more basic than those covered by the above work. SyntaxGym/CausalGym phenomena range from simple agreement patterns all the way up to long-distance dependencies.

5. Possibly as a results of the above gaps in literature coverage, the paper does not include any comparisons with top methods in this field. I think the CausalGym paper includes a pretty good overview of salient baselines and more cutting-edge methods. The current paper also includes only one very, very weak baseline (Section 6.1), which makes the results very difficult to assess. The models are not even very good at the tasks themselves (Figure 4), and so the interpretability results seem almost guaranteed to be hard to interpret.

6. I am not convinced of the value of importing the competence/performance distinction into LLM evaluation research. It's not clear that it has been a productive dichotomy within linguistics and cognitive science, and we are going to have to confront the same fraught issues here in AI. All we have is behavioral evidence from finite samples of real-world performances. We could consider trying to use this evidence to characterize capabilities that will hold for all inputs -- a challenging analytic task -- but calling this "competence" seems unlikely to clarify the nature of such results.

7. I would add that the current paper seems to use competence in an usual way relative to the linguistics literature. In linguistics, competence is one's abstract knowledge of one's language, separate from considerations that might lead to mistakes and so forth (fatigue, cognitive load, etc.). The current paper seems to want to classify models as competent or not based on prior theoretical characterizations (Section 2.3). This is very different. If we are going to adopt the  competence/performance framing, we should try to get at the underlying "knowledge" of the model rather than insisting that it conform to our prior expectations.

**Questions:**

1. BERT/RoBERTa are unusual choices for models to focus on in 2024. Is there a particular reason they were chosen?

2. I am not sure I understand the argument in the paragraph beginning on line 141. The final conclusion seems to have counterexamples in English, though. Consider "letters"/"mail" and "chairs"/"furniture". I actually am not sure what the significance of this is to the paper, though. Do my examples suggest that Figure 2 is in error? If not, why not? If so, does it matter for the paper?

3. The results in Figure 4 seem to show that the models cannot do the task, which settles the question "Are they competent?" (No.) Why not study situations in which models are successful, and try to understand what causal structures they use to be successful? Those causal structures could be argued to represent their competence.

---

### Official Review · Reviewer_4gUM · 2024-11-04

**Soundness:** 3
**Presentation:** 3
**Contribution:** 2
**Rating:** 3
**Confidence:** 3

**Summary:**

This paper proposes analyzing LLM capabilities via competence-based testing, specifically via intervening on the model’s representations via gradient-based adversarial attacks for causal probing. It is important to examine whether the linguistic competence of LLMs corresponds to that of humans, and to decouple competence from performance, since the latter might be spuriously obtained. This is especially true with modern LLMs where humans increasingly rely on them behaving in predictable and consistent ways. The CALM (competence-based analysis of LMs) framework conducts causal probing via gradient-based adversarial attacks and reports on preliminary experiments using the approach.

**Strengths:**

1. Nice approach and easy to follow exposition.
2. It makes a lot of sense to use GBIs for this purpose.
3. Promising initial results. Thorough appendix.

**Weaknesses:**

1. Lack of substance. The paper reports on some preliminary experiments, but without any clear findings. I guess it is a little surprising that BERT is more competent than RoBERTa? The preliminary analysis is conducted on language models (I wouldn't call them "large" language models) that are 6 years old, which is an eternity in our field. To really showcase the value of CALM, I would have wanted to see much more convincing experimental results. These could either be about deriving interesting and important novel insights from LLMs (ideally recent ones); or, they could be about showing why the proposed CALM framework is better than alternatives (see below).

2. Lack of comparison. Given the lack of substantial findings by utilizing the proposed approach, the paper lacks a direct empirical comparison to alternative methods to show that it is better.

Overall, I like this paper. With more work and more substance, it could be a very good submission -- but this is overall too lightweight to merit acceptance, as it stands.

**Questions:**

1. Why should I use CALM over alternatives?
2. What surprising finding have you derived from applying CALM that substantially contributes to the body of scientific knowledge?

---

### Note · Authors · 2024-11-12

I have read and agree with the venue's withdrawal policy on behalf of myself and my co-authors.